# Recognizing Vector Graphics without Rasterization

**Xinyang Jiang**[1], **Lu Liu**[2]*, **Caihua Shan**[1], **Yifei Shen**[3]*, **Xuanyi Dong**[2]*, **Dongsheng Li**[1]

[1]Microsoft Research Asia
{xinyangjiang,caihua.shan,dongsheng.li}@microsoft.com
[2]University of Technology Sydney
u.liu.cs@icloud.com,xuanyi.dxy@gmail.com
[3]The Hong Kong University of Science and Technology
yshenaw@connect.ust.hk

## Abstract

In this paper, we consider a different data format for images: vector graphics. In contrast to raster graphics which are widely used in image recognition, vector graphics can be scaled up or down into any resolution without aliasing or information loss, due to the analytic representation of the primitives in the document. Furthermore, vector graphics are able to give extra structural information on how low-level elements group together to form high level shapes or structures. These merits of graphic vectors have not been fully leveraged in existing methods. To explore this data format, we target on the fundamental recognition tasks: object localization and classification. We propose an efficient CNN-free pipeline that does not render the graphic into pixels (i.e. rasterization), and takes textual document of the vector graphics as input, called YOLaT (You Only Look at Text). YOLaT builds multi-graphs to model the structural and spatial information in vector graphics, and a dual-stream graph neural network is proposed to detect objects from the graph. Our experiments show that by directly operating on vector graphics, YOLaT outperforms raster-graphic based object detection baselines in terms of both average precision and efficiency. Code is available at https://github.com/microsoft/YOLaT-VectorGraphicsRecognition.

## 1 Introduction

Raster graphics have been commonly used for image recognition due to its easy accessibility from cameras. Most existing benchmark datasets are built upon raster graphics, from ImageNet [1] for classification to COCO [2] for object detection. However, due to its pixel-based fix-sized format, raster graphics may lead to aliasing when scaling up or down by interpolation. Fields like engineering design or graphic design require a more precise way to describe visual content without aliasing when scaling (e.g., graphic designs, mechanical drafts, floorplans, diagrams, etc), so another important image format emerges, namely *vector graphics*.

Vector graphics achieve this powerful feature by recording how the graphics are constructed or drawn, instead of the color bitmaps represented by pixel arrays defined in the raster graphics (first row in Figure 1). Specifically, vector graphics contain a set of primitives like lines, curves and circles, which is defined with parametric equations in analytic geometry and some extra attributes. As shown in Figure 1, such vector graphic is usually a document where every primitive is defined precisely and written in a line of textual command. Due to the analytic representation, with few parameters, vector graphics can represent an object at any scale or even in infinite resolution, making it potentially a lot more precise and compact image format than raster graphics. Also, instead of independent pixels,

---

*This work was done when the authors were interns at MSRA.

35th Conference on Neural Information Processing Systems (NeurIPS 2021).

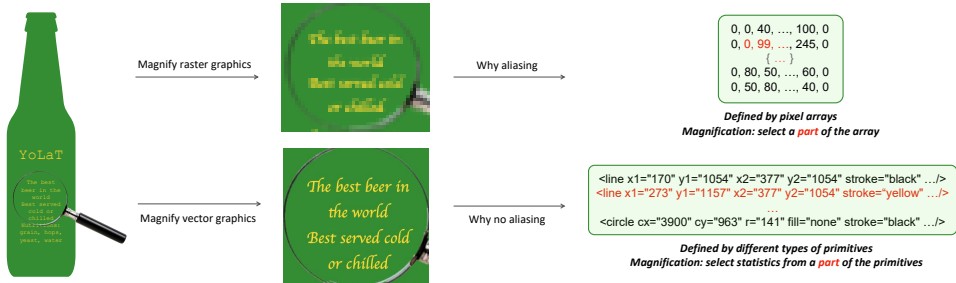

Figure 1: Difference between raster graphics (row 1) and vector graphics (row 2).

vector graphics give higher level structural information on how low-level elements like points or curves group together to form high level shapes or structures. However, this powerful and widely used data format has been rarely investigated in previous computer vision literature.

To explore this data format, this paper focuses on the fundamental recognition tasks: object localization and classification, with wide applications in vector graphics related field like automatic design audit, AI aided design, design graphics retrieval, etc. Existing raster graphics based methods [3, 4, 5, 6, 7] takes pixel arrays as input and cannot be directly applied on vector graphics. There have been attempt [8] dealing with this format by rendering vector graphics into raster graphics first. However, rendering the vector graphics into raster graphics could result in a pixel array with super resolutions (e.g., thousands by thousands), which brings extremely large memory cost, and would be inefficient or even intractable for the traditional models to process. On the other hand, rendering a lower resolution image causes substantial information loss, and the object bounding boxes obtained from a low resolution image could be imprecise when scaled back to the original resolution. Furthermore, the rendering process results in a set of independent pixels and discards the high-level structural information within the primitives. Some of this information could be critical for recognition, such as corners in a shape or contours, etc.

To address these issues, we resolve the tasks on vector graphics by introducing a model that does not need rasterization and takes the textual documents of vector graphics as input, called YOLaT(You Only Look at the Text). Instead of rendering the vector graphics into raster graphics, we propose an efficient end-to-end pipeline which predicts objects from the raw textual definitions of primitives. YOLaT first transforms different types of primitives into a unified format. Then it constructs an undirected multi-graph to model the structural and spatial information from the unified primitives. Compared to rendering to raster graphics, this transformation is able to preserve more complete information. YOLaT generates object proposals directly from the vector graphics, which produces precise object bounding boxes. Finally, a dual-stream graph neural network (GNN) specifically designed for vector graphics is proposed to classify the graph contained in each proposal, with no extra regression needed for bounding box refinement.

To evaluate our pipeline over vector graphics, we use two datasets. i.e., floorplans and diagrams and show the advantages of our method over the raster graphics based object detection baselines. Without pre-training, our method consistently outperforms raster graphics based object detection baselines, with significantly higher efficiency in terms of the number of parameters and FLOPs. Even compared with the powerful ImageNet pretrained two-stage model, YOLaT achieves comparable performance with 25 times fewer parameters and 100 times fewer FLOPs. We also show visualizations to better demonstrate why looking at the text can capture more delicate details and predicts more accurate bounding boxes.

## 2   Related Work

**Object Detection on Raster Graphics.**   Currently deep learning based object detection methods dominate the research field with the superior performance. Two-stage object detection methods first generate region proposals and classify and regress the proposals to give object predictions with a deep convolutional networks. R-CNN [9] and Fast-RCNN [3] use *selective search* for proposal generations. Faster-RCNN [10] speeds up the proposal generation by introducing a region proposal network. He et al. [11] proposed Mask-RCNN, adding a segmentation branch to the detection model

for instance segmentation. To train a more translation-variant backbone, Dai et al. [12] proposed F-RCN – a new prediction head with position-sensitive score maps.

Most two-stage object detection methods have large computation overhead of the proposal generation process, and require running a classification and regression sub-network on all the region proposals. One-stage object detection methods tackle this challenge by removing the proposal generation process and directly predict the object bounding boxes in an end-to-end fashion. Anchor-based methods like SSD [4], YOLO series [5, 13, 14, 15], RetinaNet [6] densely tile anchor boxes over the image and conduct classification and bounding box coordinate refinement on each anchor box. Recently, anchor-free methods like CornerNet [16], CenterNet[7], FCOS [17] propose to directly find object without presets anchors.

**Graph Neural Networks.** GNN has become a powerful tool for machine learning on graphs. It computes a state for each node in a graph, and iteratively updates the node states according to its neighbors. Spectral approaches [18, 19] define a convolution operation in the Fourier domain. Spatial approaches [20, 19, 21] define convolutions directly on the graph. EdgeConv [22] applies GNN model for classification on 3D Cloud by taking the state difference between neighboring nodes as the input of the aggregation function. [23] further applies EdgeConv to the object detection task on 3D cloud data by integrating the GNN backbone into an anchor-free detection framework. The closest GNN model to our YOLaT is EdgeConv but YOLaT has extra upgrade specifically designed for vector graphics, including edge attributes, faster inference on densely connected edges, and dual-stream structure for multi-graph.

**Online Sketch and Handwriting Recognition.** Online handwriting and drawing recognition [24, 25] handles a data form that very similar to vector graphics, which contains a sequence of discrete points. Most of these methods use sequential models to handle this problem. For example, [26] proposes to convert the point sequences to a sequence of Bézier Curves and use a LSTM for sequential modeling. Compare to online handwriting, vector graphics contain more types of un-ordered shapes with more attributes and properties other than polylines, and hence need more general and non-sequential method.

**Vector Graphics Related Application.** One of the most common application for vector graphics is design, such as architecture, graphic design, etc. Several methods in architecture drawing recognition propose to represent symbols in a floor-plan as graphs, and use rule-based graph matching method to classify and localize symbols, such as visibility graph [27] and attributed relational graph [28, 29]. In this paper, we propose a novel scheme that directly construct graph from vector graphics based on Bézier Curve, and the object detection is conducted based on the prediction of GNN. Recent years, a few works develop deep learning based methods to automatically generate vector graphics for computer aided design or converting raster graphics to vector graphcis (i.e. vectorization) [30, 31, 32, 33, 34, 35], while to the best of knowledge, our paper is the first to focus on recognition task on vector graphics. Koch et al. [36] proposes a large 3D model dataset containing analytic representations, but it lacks semantic labeling to train recognition model.

## 3 Detection Model

In this paper, we study the problem of object detection leveraging the definitions of the vector graphics without rendering them. Here, we define the task as object localization and object classification. Specifically, the model needs to predict a set of bounding box coordinates as well as the category of the object within the bounding boxes.

In this section, we describe our proposed YOLaT, which is an end-to-end efficient pipeline taking the raw definitions of the vector graphics as the input without further rendering the graphics. Figure 2 shows the overall pipeline of YOLaT. We convert the primitives like lines and curves as a universal format of Bézier curves. Based on the Bézier curves, un-directed multi-graphs are constructed to model both spatial and structural relationships among the key-points within a primitive and among different primitives. More details on how we build the graphs can be found in Section 3.1. To fully explore the vector graphics based on the multi-graph, we propose a dual-stream GNN for graph feature extraction and classification. Section 3.2 shows the detailed design of the proposed dual-stream GNN. Compared to the complex prediction head commonly used in the object detection for raster graphics, YOLaT generates precise proposal bounding boxes directly from high resolution vector graphics. Hence each sub-graph in the proposals are fed into the dual-stream GNN classifier

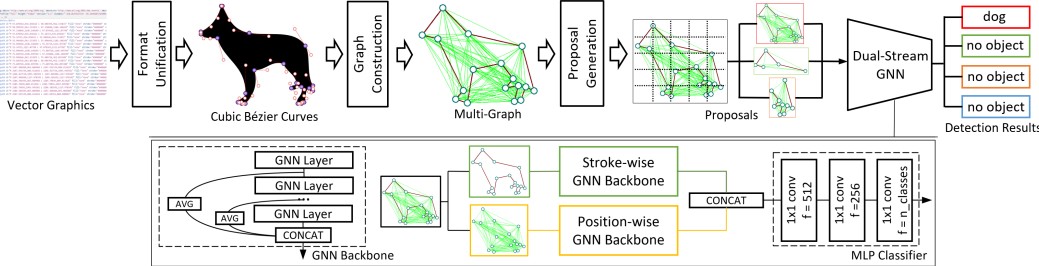

Figure 2: The overall pipeline of the proposed method.

without further correction of the box coordinates. We show how to get the potential bounding boxes and predict their objectiveness and category in Section 3.3.

## 3.1 Graph Construction

**Universal formats of curves.** Compared to the raster graphics represented by pixel arrays, vector graphics have more precise representations and no loss of quality and aliasing when resizing. The vector graphics consists of primitives defined by textual commands described in parametric equations, such as lines, curves, polygons and other shapes.Different primitives are described with different parametric equations. Here, like pixel in raster graphics, we want to find a unified way to describe all types of primitives. We chose Bézier Curve due to its generality and capability of modeling different shapes and curves. Bézier Curve is defined by a set of control points, $\{\boldsymbol{p}_0, ..., \boldsymbol{p}_n\}$, where $n$ is the order of the curve and $\boldsymbol{p}_{i\in[n+1]}$ is a 2-d vector for the coordinates of point $i$. The first point and the last point are the end points of a curve while the rest of the control points usually do not sit on the curve and provide side information instead, such as directional information and curvature statistics of the curve from $\boldsymbol{p}_0$ to $\boldsymbol{p}_n$. We chose cubic Bézier Curve where $n = 3$ for the balance between modeling capability and computational complexity. Formally, the cubic Bézier curve $\boldsymbol{B}$ is defined as:

$$\boldsymbol{B}(t) = (1-t)^3\boldsymbol{p}_0 + 3(1-t)^2t\boldsymbol{p}_1 + 3(1-t)t^2\boldsymbol{p}_2 + t^3\boldsymbol{p}_3, 0 \leq t \leq 1 \tag{1}$$

where $\boldsymbol{B}(t)$ defines the position of a specific point at the scale rate of $t$ on the curve from $\boldsymbol{p}_0$ to $\boldsymbol{p}_3$. Next, we introduce our graph construction based on a set of Bézier Curves.

**Nodes.** To improve efficiency, we only include the points from the set of start points and end points, denoted by $\mathbb{P}$, into the collections of nodes on graphs. The rest of the control points will serve as the edge attributes as defined in the following paragraph. For a point $\boldsymbol{p}$, the attributes $\boldsymbol{x}$ of the corresponding node include the coordinates of the point, the RGB color value $\boldsymbol{c}$ and stroke width $w$:

$$\boldsymbol{x} = \text{concat}(\boldsymbol{p}^x, \boldsymbol{p}^y, \boldsymbol{c}, w), \boldsymbol{p} \in \mathbb{P} \tag{2}$$

where $\boldsymbol{p}^x$ and $\boldsymbol{p}^y$ denote the coordinate value of the point $\boldsymbol{p}$ along the x axis and y axis respectively. These information is defined in the vector graphic documentation.

**Edges.** We design the graph as a multi-graph containing two sets of edges, namely the stroke-wise edges and the position-wise edges. These two types of edges capture the node relationships from different perspectives.

*Stroke-wise edges* capture the connections defined by the stroke in the vector graphics, which refers to the actual stroke drawn between the start and end point of each Bézier curve. This type of connections represents the structures and layouts of the objects in the vector graphics. Thus, an edge is built in-between two nodes if there is a Bézier Curve linking them:

$$\mathcal{E}_s = \{(v_i, v_j) : (v_i, v_j) \in \mathbb{S}\} \tag{3}$$

where $\mathbb{S}$ denotes the set of tuples containing the start point $v_i$ and end point $v_j$ of a Bézier Curve.

Other than the connections between start and end points, other attributes of a cubic Bézier curve like curvature or other appearance are described by the off-curve control points. We use the coordinates of these off-curve control points as the attributes of the stroke-wise edges:

$$\boldsymbol{x}^e = \text{concat}(\boldsymbol{p}^x, \boldsymbol{p}^y), \boldsymbol{p} \notin \mathbb{P} \tag{4}$$

The stroke-wise edges only model the long-term structural connection between the vertices based on strokes, which is irrelevant to the spatial vicinity. To further capture the spatial relationship between nodes, we generate another set of edges, called *position-wise edges*. Specifically, the position-wise edges are defined as the dense connections among nodes within a regional cluster $\mathbb{C}_k$:

$$\mathcal{E}_p = \{(v_i, v_j) : v_i, v_j \in \mathbb{C}_k\}, k \in \{1, 2, ..., m\}, \tag{5}$$

A regional cluster is a set of nodes close to each other spatially, which can by obtained in different ways. In our implementation, we obtain regional cluster in three steps. First, given our graph representation of a vector graphic, we obtain all the connected components in the graph, based on the stroke-wise edges $\mathcal{E}_s$. Secondly, for each pair of connected components, obtain their expanded minimum bounding rectangles and the overlapping area of the rectangles. If the expanded area of two connected components overlap, they are spatially close and are merged to be one regional cluster $\mathbb{C}_k$. The expand length is a hyper-parameter.

## 3.2 Feature Extraction with Dual-stream GNN

In the previous section, we introduced how to generate graphs, including building nodes, two sets of edges and attributes. To analyze the proposed multi-graph, YOLaT applies a GNN network specifically designed for the graph built from vector graphics. Since the proposed graph is defined by two sets of edges at hand, YOLaT uses a dual-stream GNN structure where a specific GNN branch is designed to update node representations based on each type of edges. The node representations extracted by the dual-stream GNN are able to leverage the spatial and structural information in vector graphics, and can better guide the following head for the downstream tasks.

In the following section, we first elaborate on the details of both streams in our GNN. Then we introduce how to get the representation of a specific region by leveraging multi-step node representations propagation and representation fusion.

**Stroke-wise Stream.** For the graphs with stroke-wise edges, inspired from [37], this GNN takes the input of the concatenation of a node representation, the difference of the representation to its neighbor node, and the attributes on the edge. At time step $t + 1$, the representations $\mathbf{h}_i^{t+1}$ for a node $i$ is updated as follows:

$$\mathbf{h}_i^{t+1} = f^l(\mathbf{h}_i^t) + \frac{1}{|\mathcal{N}_i^s|} \sum_{j \in \mathcal{N}_i} f^s(\text{concat}(\mathbf{h}_i^t, \mathbf{h}_j^t - \mathbf{h}_i^t, \mathbf{x}_{ij}^e)), \tag{6}$$

where the initialization is calculated as Equation 2, i.e., $\boldsymbol{h}^0 = \boldsymbol{x}$. $\mathbf{x}_{ij}^e$ denotes the attributes on the stroke-wise edge between node $i$ and node $j$ as defined in Equation 4. $f^l$ is a linear transformation function. $\mathcal{N}_i^s$ denotes set of nodes adjacent to the $i$-th node in terms of stroke-wise edges in the graph, and $f^s$ denotes a transformation function which consists of a linear transformation, a ReLU activation function [38] and a batch normalization layer [39] in our implementation.

**Position-wise Stream.** Since the position-wise edges are constructed densely as described in Section 3.1, the number of position-wise edges is significantly larger than that of the stroke-wise edges. To reduce the computational cost, we design a simpler GNN model for graphs with position-wise edges. At time step $t+1$, the model only takes the representation of a node and the node representation $\mathbf{z}_i^{t+1}$ is updated by considering the neighboring transformed representation:

$$\mathbf{z}_i^{t+1} = \frac{1}{|\mathcal{N}_i^p|} \sum_{j \in \mathcal{N}_i^p \cup \{i\}} f^p(\mathbf{z}_j^t), \tag{7}$$

where $\mathcal{N}_i^p$ denotes the neighbors of node $i$ defined by the positional edges (to maintain the information from $v_i$ a self-loop for each node is added). $f^p$ is a transformation function with the same structure but untied parameters as $f^s$. The initialization of the node representation is also calculated as Equation 2, i.e., $\mathbf{z}^0 = \boldsymbol{x}$. Compared to stroke-wise edge, the computation complexity of $f^p$ can be reduced significantly, because the updates of the nodes in the same regional cluster $\mathbb{C}_k$ only needs to be computed once. More details of the efficient implementation for the GNN can be found in Section 4.

**Representation Fusion.** The representation of a specific region in vector graphics is based on the cluster of nodes representations located within this region. Specifically, given the node representations

refined by the proposed dual-stream GNN, we average the node representations over the nodes inside this region $\mathcal{V}^r$ to get a region representation and concatenate the region representations for $\mathcal{T}$ steps:

$$\mathbf{r} = \text{concat}(\mathbf{r}_s^0, \mathbf{r}_s^1, ..., \mathbf{r}_s^{\mathcal{T}}, \mathbf{r}_p^0, \mathbf{r}_p^1, ..., \mathbf{r}_p^{\mathcal{T}}), \tag{8}$$

$$\mathbf{r}_s^t = \frac{1}{|\mathcal{V}^r|} \sum_i \mathbf{h}_i^t, i \in \mathcal{V}^r, \qquad \mathbf{r}_p^t = \frac{1}{|\mathcal{V}^r|} \sum_i \mathbf{z}_i^t, i \in \mathcal{V}^r, \tag{9}$$

where $\mathbf{r}$ denotes the fused representation of a specific region and $\mathbf{r}_s^t$, $\mathbf{r}_p^t$ denote the region representation at time step $t$ from the graph with stroke-wise edges and position-wise edges, respectively.

### 3.3  Prediction and Loss

Here we propose a vector graphics based proposal generation method. Given a vector graphic, we first evenly slices each regional clusters $\mathbb{C}_k$ into grids. Then, we permute all vertex pairs on the grid mesh, each of which forms the top-left and bottom-right points of a rectangle region. The nodes, edges and corresponding primitives within each rectangle region forms a proposed object, whose minimum bounding rectangle is the bounding box of the proposal. Note that proposals with size larger than a threshold is filtered. Compared to generating proposals on raster graphics, YOLaT produces much fewer negative samples, and operates at highest resolution to directly produce tightest bounding boxes around the proposed object. Hence, YOLaT requires no extra regression branch for bounding box refinement.

For each proposal, a proposal $\hat{B}$'s representation $\boldsymbol{r}$ is obtained by the representation fusion strategy as described in previous section, which is then fed into a multi-layer perception to predict object category. During training, we only optimize the cross-entropy loss over the prediction and $\hat{B}$'s ground truth label $y$. In each image, for each ground truth object box $B_i$, its label is $y_i$. We set $y$ the same as that of the ground truth $B_i$, which has the largest Intersection over Union (IoU) with the proposal $\hat{B}$. If the largest IoU is below a threshold $\alpha$, we regard this proposal as "no object" and set its ground truth label as the total number of classes $C$ (the class index is from 0). We minimize the cross entropy loss of each proposal:

$$\min - \log \Pr(y|\hat{B}) = \min - \log \Pr(y|\boldsymbol{r}), \tag{10}$$

$$y = \begin{cases} \arg\max_{y_i \in \mathcal{Y}} \text{IoU}(\hat{B}, B_i) & \text{if } \max(\text{IoU}(\hat{B}, B_i)) >= \alpha \\ C & \text{else} \end{cases}, \tag{11}$$

where $\mathcal{Y}$ is the set of all ground truth labels for the boxes $\{B_i\}$.

During evaluation, we regard the probability of the classification as the confidence level for this proposal and select the bounding boxes with the confidence level above a set threshold as the predictions.

## 4  Experiments

### 4.1  Implementation Details

**Architecture.** In the model used in our main results comparison, we build a two-layer GNN for both position-wise stream and stroke-wise stream with dimension of all the hidden node representations set to $64$. We observe no significant performance improvement with deeper GNN due to the over-smoothing effect. In our graph, the number of position-wise edges is quite large due to its full connectivity within each regional cluster. To speed up the inference, we first pre-compute the transformation function $f^p$ on each node. Then for each regional cluster, we aggregate the obtained node representations with mean-pooling. The aggregated representation for each regional cluster is then assigned to each node in the cluster as their new node representation. In this way, each node only requires one transformation operation and one mean-pooling operation. Furthermore, since the fully connected graph constructed by position-wise edges could cause severe over-smoothing problem, after run $f^p$ on each node, the mean aggregate operation is only conducted on the last layer of GNN. We use a three-layer MLP as classifier, where the dimension of middle layer output is $512$ and $256$.

**Graph Construction and Proposal Generation.** Our experiments use a widely used vector graphics standards called Scalable Vector Graphics (SVG). All the primitives in SVG are first converted to cubic Bézier Curves. A circle is split equally into four parts and then each part is converted into Bézier

Table 1: Performance comparison on the floorplan dataset.

| Methods | Pretrained | AP$_{50}$ (%) | AP$_{75}$ (%) | mAP (%) | Inference time (ms) | Params(M) | GFLOPs |
|---|---|---|---|---|---|---|---|
| Yolov3-tiny | ✗ | 75.23 | 60.97 | 53.24 | **1.2** | 8.7 | 13.0 |
| Yolov3 | ✗ | 88.24 | 80.44 | 72.98 | 8.2 | 61.6 | 155.2 |
| Yolov3-spp | ✗ | 87.38 | 79.66 | 71.61 | 8.3 | 62.7 | 156.1 |
| Yolov4 | ✗ | 93.04 | 87.48 | 79.59 | 11.7 | 70.3 | 165.5 |
| faster-rcnn-R18-FPN | ✗ | 80.91 | 71.48 | 67.32 | 58.7 | 28.4 | 126.8 |
| faster-rcnn-R34-FPN | ✗ | 80.50 | 72.18 | 65.89 | 61.9 | 38.5 | 157.3 |
| faster-rcnn-R50-FPN | ✗ | 80.31 | 73.28 | 66.53 | 73.3 | 41.4 | 165.7 |
| retinanet-R50-FPN | ✗ | 87.50 | 82.91 | 79.18 | 79.2 | 38.0 | 189.2 |
| **YOLaT (Ours)** | ✗ | **98.83** | **94.65** | **90.59** | 1.3 | **1.6** | **1.5** |
| faster-rcnn-R50-FPN | ✓ | 98.04 | 95.23 | 90.25 | 71.2 | 41.4 | 165.6 |
| Yolov3 | ✓ | 74.61 | 60.33 | 53.76 | 8.2 | 61.6 | 155.2 |

curves. We also split curves at the intersection into multiple sub-curves to model delicate differences. For proposal generation, each region cluster is slices into a grid with 10 columns and 10 rows.

**Training.** We use Adam optimizer with a learning rate of 0.0025 and a batch size of 16. For data augmentation, we randomly translate and scale the vector graphics by at most 10% of the image width and height, and the transformed vector graphics are further rotated by a random angle. The model is trained for 200 epochs from scratch which takes around 2 hours on a Nvidia V100 graphic card.

## 4.2 Datasets

We use SESYD, which is a public database containing different types of vector graphic documents, with the corresponding object detection groundtruth, produced using the 3gT system[1]. Our experiments use the floorplans and diagrams collections.

**Floorplans.** This dataset includes vector graphics for floorplans. It contains 1,000 images with totally 28,065 objects in 16 categories, e.g., armchair, tables and windows. The images are evenly divided into 10 layouts. We divide half of the layouts as the training data and the other half for validation and test. The ratio of the validation and test data is 1:9.

**Diagrams.** This dataset includes vector graphics for diagrams. It contains 1,000 images with totally 1,4100 objects in 21 categories, e.g., diode and resistor. There are 10 layouts and 100 images for each layout. Note that scale variance of different objects is huge in this dataset. For example, a resistor is often much smaller compared to a transistor. We divide the training, validation and test set in a way that objects from the same category are included in both training and testing set. Thus, the dataset is split as 600, 41 and 359 images for training, validation and test stage.

## 4.3 Evaluation Metric

We evaluate the models in terms of both accuracy and efficiency. For accuracy, we use AP$_{50}$, AP$_{75}$ and mAP, where AP$_*$ represents the average precision with the intersection over union (IOU) threshold for counting as detected as 50%, and 75%. mAP is the mean of the average precision for the IOU threshold between 0.50 and 0.95. We also evaluate the efficiency because of the real-world requirements for real-time object detection. Specifically, we use GFLOPs (Giga (one billion) Floating point operations) and inference time for model efficiency and we also report the number of parameters to meet the scenarios of limited resources. The inference time is evaluated on a Nvidia V100.

## 4.4 Comparison to Baselines

We compare YOLaT with two types of object detection methods: one-stage methods, i.e., Yolov3 [14], Yolov4 [15, 40] and its variants, RetinaNet [6], and two-stage methods, i.e., faster-rcnn with Pyramid Network (FPN) [41] and its variants. For Yolov3, the -tiny variant is a smaller model and the -spp uses Spatial Pyramid Pooling. For Yolov4, we use a scaled Yolov4 [40] with slightly more parameters and potentially much better performance called Yolov4-P5. The faster-rcnn-R*-FPN model series

---

[1]http://mathieu.delalandre.free.fr/projects/sesyd/

Table 2: Performance comparison on the diagram dataset.

| Methods | Pretrained | AP$_{50}$ (%) | AP$_{75}$ (%) | mAP (%) | Inference time (ms) | Params (M) | GFLOPs |
|---|---|---|---|---|---|---|---|
| Yolov3-tiny | ✗ | 88.40 | 79.53 | 71.42 | 3.6 | 8.7 | 13.0 |
| Yolov3 | ✗ | 89.69 | 81.38 | 78.20 | 10.9 | 61.6 | 155.2 |
| Yolov3-spp | ✗ | 90.29 | 84.51 | 78.68 | 10.8 | 62.7 | 156.1 |
| Yolov4 | ✗ | 88.71 | 84.65 | 76.28 | 11.1 | 70.3 | 165.5 |
| faster-rcnn-R18-FPN | ✗ | 92.79 | 89.10 | 85.89 | 34.4 | 28.4 | 121.1 |
| faster-rcnn-R34-FPN | ✗ | 90.47 | 88.74 | 85.21 | 36.0 | 38.5 | 150.0 |
| faster-rcnn-R50-FPN | ✗ | 91.88 | 90.25 | 84.65 | 44.9 | 41.7 | 158.0 |
| retinanet-R50-FPN | ✗ | 91.33 | 83.17 | 82.79 | 47.9 | 38.0 | 179.5 |
| **YOLaT (Ours)** | ✗ | **96.63** | **94.89** | **89.67** | **2.1** | **1.6** | **2.9** |
| faster-rcnn-R50-FPN | ✓ | 95.24 | 93.57 | 90.76 | 40.0 | 41.7 | 157.9 |
| Yolov3 | ✓ | 90.11 | 84.68 | 79.55 | 8.2 | 61.6 | 155.2 |

Table 3: Ablation study and variant analysis on the floorplan dataset.

(a) Ablation study on graph construction.

| Methods | AP$_{50}$(%) | AP$_{75}$(%) | mAP(%) |
|---|---|---|---|
| YOLaT | 98.83 | 94.65 | 90.59 |
| w/o $\mathcal{E}_p$ | 95.81 | 91.03 | 87.17 |
| w/o $\mathcal{E}_s$ | 91.57 | 91.22 | 86.00 |
| w/o $\mathbf{x}_{ij}^e$ | 94.57 | 90.76 | 86.25 |

(b) Ablation study and variant analysis on GNN model.

| | Methods | AP$_{50}$(%) | AP$_{75}$(%) | mAP(%) |
|---|---|---|---|---|
| stroke-wise stream | w/o $h_i^t$ | 96.83 | 93.19 | 88.40 |
| | w/o $\mathbf{h}_j^t - \mathbf{h}_i^t$ | 95.87 | 92.90 | 87.83 |
| position-wise stream | early aggregate | 95.82 | 91.64 | 87.90 |
| | with $\mathbf{h}_j^t - \mathbf{h}_i^t$ | 98.67 | 94.46 | 90.39 |
| aggregation function | GCN | 90.36 | 88.02 | 83.32 |
| | GAT | 91.20 | 89.46 | 83.92 |
| | GraphSage | 92.70 | 91.17 | 85.26 |

use backbones of different scales, with ResNet18 [42], ResNet34, ResNet50 for R18, R34, R50, respectively.

We choose these baselines because they are the most popular methods in object detection. On both datasets, YOLaT outperforms all baselines without pretraining on ImageNet in terms of precision and efficiency as shown in Table 1 and Table 2. We also include a baseline, i.e., faster-rcnn-R50-FPN which is pretrained on ImageNet, YOLaT shows competitive precision with around $100\times$ less FLOPs and around $25\times$ less model parameters. We also train Yolov3 with ImageNet pretrained backbone, but do not observe performance improvement. We conduct 3 rounds of experiments with different random seeds and the standard error in terms of AP$_{50}$ is 0.0003 on floorplan and 0.0008 on diagram.

For Yolov3, we use the implementation of ultralytics[2] [43]. For Yolov4 we use the official pytorch implementation of Scaled Yolov4[3] [40]. Note that both the Yolov3 and Yolov4 implementation shows superior performance on COCO [2] when without ImageNet pretraining. For faster-rcnn and retina-net, we use the Detectron2 [44] library[4]. For the non-pretrained model, we use the strategies of replacing Batch normalization to Group Normalization following [45] to improve the performance.

**Broader Impact.** Our YOLaT model may present a promising solution for applications that have the input of vector graphics. Any deployment of the proposed model however should be preceded by an analysis of the potential biases captured by the dataset sources used for training and the correction of any such undesirable biases captured by the pre-trained backbones and model.

## 4.5 Ablation Study and Variants Analysis

**Graph Construction.** We analyze the effectiveness of the Bézier based graph construction method in YOLaT on SYSED-Floorplans dataset. Table 3a shows the results of the ablation study on position-wise edges $\mathcal{E}_p$ (as defined in Eq. 7), stroke-wise edges $\mathcal{E}_s$ (as defined in Eq. 6) and edge attributes $\mathbf{x}_{ij}^e$ (as defined in Eq. 2). The ablation of these components show a significant drop in precision.

**Dual-Stream GNN.** As show in Table 3b, we conduct several experiments to analyze the effectiveness of our dual-stream GNN that is specifically designed for vector graphics recognition. We did the

---

[2]https://github.com/ultralytics/yolov3
[3]https://github.com/WongKinYiu/ScaledYOLOv4
[4]https://github.com/facebookresearch/detectron2

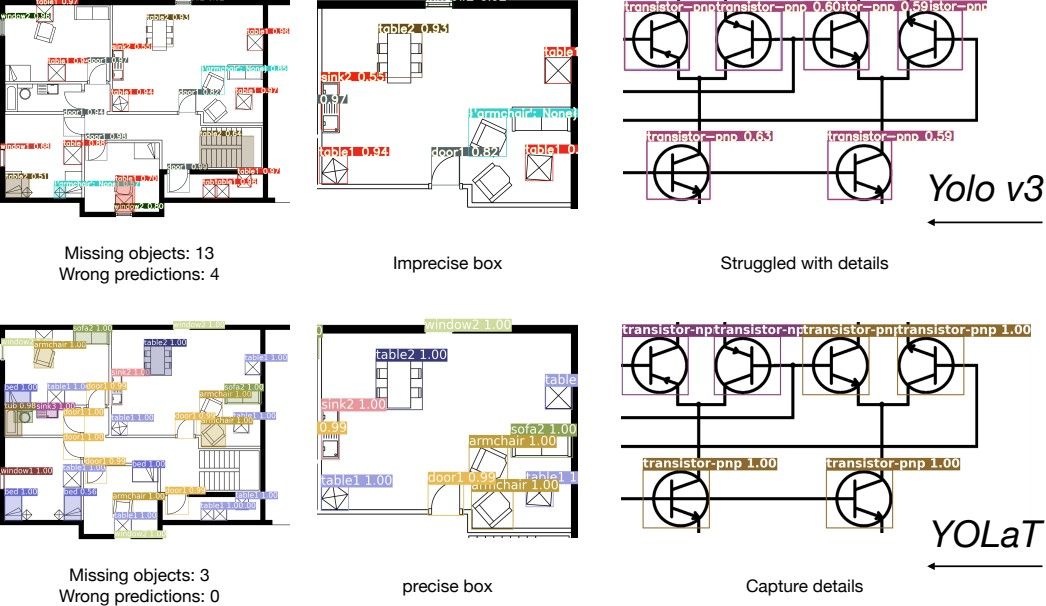

Figure 3: Visualizations of Yolov3 (upper line) and YOLaT (lower line) show that 1) [*Left*] Yolov3 has more missing objects (shaded boxes in YOLaT figure) and wrong predictions (shaded boxes in Yolov3 figure). 2) [*Mid*] The prediction boxes of YOLaT are tighter and more accurate. 3) [*Right*] Yolov3 can not distinguish the details of transistors, e.g., the direction of the arrows, leading to wrong predictions.

ablation of YOLaT without the input $\boldsymbol{h}_i^t$ and $\boldsymbol{h}_j^t - \boldsymbol{h}_i^t$. For position-wise stream, feature aggregation for position-wise edges is conducted on every layer in GNN, instead of only last layer. the experiment results show that early aggregation hurts the performance, due to the over-smoothing caused by fast message passing along the fully connected edges. Due to the high computation complexity of aggregation function on fully connected position-wise edges, YOLaT discards neighboring feature difference in $f^p$. This experiment shows that there is no obvious performance improvement by adding neighboring feature difference on $f^p$. Meanwhile, this method significantly increases the computation complexity by and increase GFLOPs by almost $60\%$ from $1.5$ to $2.4$. The last three rows of Figure 3b shows the performance comparison between YOLaT and some other popular GNN aggregation methods. In this experiments, we replace our proposed aggregation function with the aggregation functions in GCN [19], GAT [21] and GraphSage [20]. Since some of these methods do not directly support edge attributes, similar to our dual-stream GNN, we treat it as extra dimensions of features of a pair of adjacent nodes. The experiment shows that our GNN outperforms existing GNN methods, which further verifies the effectiveness of our vector graphics specific design.

## 4.6 Visualizations

We visualize the detection results for Yolov3 and YOLaT as in Figure 3. The prediction results in first two columns show that the bounding box predicted by Yolo is imprecise while the bounding box predicted by YOLaT is precise and sits exactly at the border of every object. For example, both models generate a bounding box for the table in the middle of the figure, while YOLaT outputs tighter box for the object border. This is because YOLaT directly looks at the text and leverages the information of where the positions of the curves are, while Yolo only leverages the lower resolution pixel arrays rendered from the text. The imprecise predictions can affect the performance for higher standard detection which is reflected as the AP with higher IOU. This is why the gap between Yolo and YOLaT is bigger for mAP compared to that for AP50 as shown in Table 1. Also, Yolo gives more undetected cases under a strict threshold, such as the armchair and sofa as in Figure 3.

The third column on Figure 3 shows that Yolov3 fails to distinguish the object details (the direction of arrows in different types of transistors) due to the limited resolution of raster graphics, while by

directly operating on the vector graphics with each primitive precisely described by textual command, YOLaT is able to capture the details at very small scale.

## 5 Conclusions

We propose an efficient CNN-free pipeline does not need rasterization called YOLaT(You Only Look at Text). YOLaT builds a unified representations for all primitives in a vector graphic with un-directed multi-graph and detect objects with a dual-stream GNN specifically designed for vector graphics. The experiments show that YOLaT outperforms both one-stage and two-stage deep learning methods with much better efficiency. Our work provides a new direction for recognition on vector graphics, and is able to draw more researchers' attention on exploring the merits of vector graphics. In the future, there is much work to further improve YOLaT and recognition on vector graphics in general, such as leveraging both vector graphic and raster graphic based methods, building a GNN model for vector graphics that supports deeper structure, large vector graphics dataset to support backbone pre-training.

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
