# Appendix for Recognizing Vector Graphics without Rasterization

**Xinyang Jiang**[1], **Lu Liu**[2]*, **Caihua Shan**[1], **Yifei Shen**[3]*, **Xuanyi Dong**[2]*, **Dongsheng Li**[1]

[1]Microsoft Research Asia
{xinyangjiang,caihua.shan,dongsheng.li}@microsoft.com,
[2]University of Technology Sydney
u.liu.cs@icloud.com,xuanyi.dxy@gmail.com
[3]The Hong Kong University of Science and Technology
yshenaw@connect.ust.hk

## A    Hyper-parameters

**Number of GNN Layers.**

Table 1 shows how the number of layers in GNN influence the model performance. We observe that even a GNN with few layers can achieve satisfactory performance. Increasing layers in our GNN also does not bring very significant performance improvement or even hurts the performance due to over-smoothing.

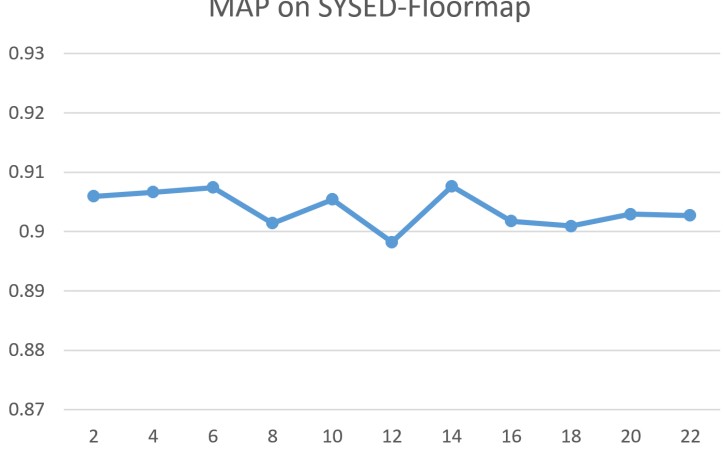

Figure 1: AP50 comparison with different layers of GNN on floorplan dataset.

**Number of Proposals.**

We analyze how density of generated proposals affects the performance by split the multi-graph spatially into mesh grid with different number of strides. More strides in grid and more proposals bring higher detection performance. When the number of strides is over 10, the improvement is insignificant compared to the computational cost.

---

*Works done when authors interned in MSRA

35th Conference on Neural Information Processing Systems (NeurIPS 2021).

Table 1: The performance comparison with different number of strides in the mesh grid for proposal generation on diagram dataset

| Number of Strides | $AP_{50}(\%)$ | $AP_{70}(\%)$ | mAP(%) | Average Number of Proposals | GFLOPs |
|---|---|---|---|---|---|
| 3 | 89.35 | 85.47 | 82.53 | 144.8 | 0.4 |
| 5 | 94.34 | 90.53 | 85.94 | 327.4 | 1.0 |
| 10 | 96.63 | 94.89 | 89.67 | 959.9 | 2.9 |
| 15 | 96.96 | 95.01 | 89.64 | 1394.4 | 4.2 |
| 20 | 97.02 | 95.46 | 89.84 | 1667.8 | 5.0 |

## B  Bézier curves Conversion

A vector graphic file contains multiple lines of textual commands, and each line defines a specific primitive/shape in the image, including the shape type (e.g., circle, line, Bézier curve, etc) and its associated parameters (e.g., start/end/center point coordinates).

After parsing the shape category and parameters from the command, each shape (or a part of the shape) can be converted into a Bézier curve with a closed-formed expression. Here we take circle as an example. A circle is split into four equal sections, ie., left-up quarter, left-bottom quarter, right-up quarter and right-bottom quarter, and each is converted to a Bézier curve. For a circle centered at the origin with radius 1, and the right-up quarter start at (0, 1) and end at (1, 0), the control points of the corresponding Bézier curve can be obtained by:

$$P_0 = (0,1), P_1 = (c,1), P_2 = (1,c), P_3 = (1,0), c = 0.551915024494 \tag{1}$$

which gives a maximum radial error to the original circle less than 0.02

Due to page limit, we only briefly introduce it in Section 5.1 Line 237-241. More details can be found in the source code in the supplemental materials and will be added into the appendix.

## C  Visualizations

We provide more visualizations as in Figure 2 to shed lights on how our models do prediction.

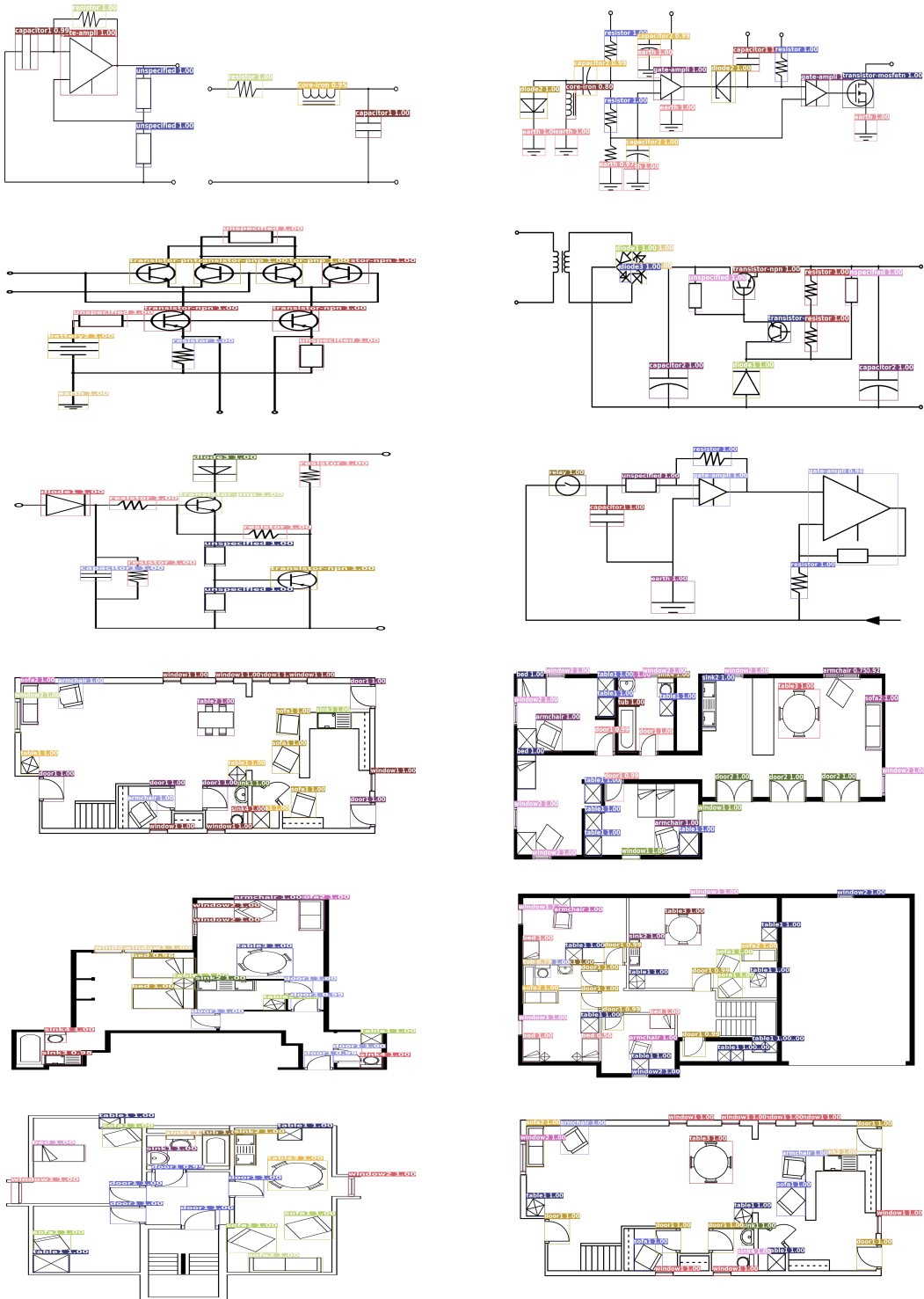

Figure 2: Visualization of predictions by YOLaT.