# OpenReview forum: "Recognizing Vector Graphics without Rasterization"
_NeurIPS.cc/2021/Conference — NeurIPS 2021 Poster_

### Official Review · Reviewer_qaZw · 2021-07-05

**Rating:** 6
**Confidence:** 4

**Summary:**

Update at the end of the discussion session:
Discussion among reviewers and with the authors made me increase my score by a total of 2 points.

The paper presents a graph-neural network based approach to detecting objects in vector graphics. The paper works directly on the vector graphics representation by first normalizing the vector graphics to bezier splines and then building a graph representation of the vector graphics. Object detection happens directly on this graph representation. Experiments show a comparison to standard computer vision models.

**Ethical Concerns:**

no concerns

**Limitations And Societal Impact:**

I personally don't think the evaluation in the paper is sufficient to understand limitations. Experiments on two datasets of 1000 images of unknown difficulty don't give a good idea of what the limitations of the method are.

**Main Review:**

## Further updates after more discussion:

increased score to "above decision threshold" assuming:
 - title is changed
 - clarification about baselines not being strong baselines

## Updated review following on discussion with other reviewers and authors:

Review up'd by 2 points because:
positive:
 - it's an important topic
 - the model is interesting

negative:
 - experiment baselines are lacking
 - online handwriting / drawing recognition work is relevant
 - the title is misleading

The paper presents an application of GNNs to detecting objects in vector graphics.


REFERENCES AND RELATED WORK
While working on vector graphics is not a very common research domain at the moment, the authors do not refer to the related work that has been looking at vector graphics, e.g.

[1] A Learned Representation for Scalable Vector Graphics https://arxiv.org/abs/1904.02632

further, the entire domain of online handwriting and drawing recognition also handles vector graphics, in in particular is often using a very similar representation:

[2] SketchRNN  - https://arxiv.org/abs/1704.03477
[3] CoSE: Compositional Stroke Embeddings - NeurIPS 2020
[4] Fast Multi-language LSTM-based Online Handwriting Recognition - is using a Bezier representation - similar to what is proposed here for handling (vector graphics of) handwriting
[5] BézierSketch: A generative model for scalable vector sketches - ECCV 2020 https://arxiv.org/abs/2007.02190

Further, in the domain of architectural drawings there was:
[5] PlanIT: Planning and Instantiating Indoor Scenes with Relation Graph and Spatial Prior Networks - https://dl.acm.org/doi/pdf/10.1145/3306346.3322941


Beyond that, I feel that the paper title and representation of the work is misleading: Just looking at the text would in my mind imagine just looking at the raw SVG file - however, as I am reading the paper it becomes clear that there is a lot of specialized preprocessing to normalize the vector graphics and create the graph. So, "only look at text" - is clearly not a good representation of the paper.


TECHNICAL
in section 4, is a description of how the graph is build which includes a step in which the input drawing is converted to bezier curves. It is not clear however, how this would work in the case of a very long polyline or e.g. for a circle which are both not easily represented as a Bezier curve. It feels like for this the shape need to be split into multiple bezier splines - but it is unclear from the paper how that is done.


EXPERIMENTAL EVALUATION
 - it is not surprising that a computer vision model that was pre-trained on very different data and then fine tuned on ~600 images doesn't perform particularly well on objects that are mostly defined by their outlines rather than by their textures.


MINOR ASPECTS
 - lots of abbreviations are badly capitalized throughout the paper, e.g. "Yolov3" instead of YOLOv3, "faster-rcnn",...
 - similarly the references are badly formatted
 - Figure 2 uses an image of a dog as example - I feel it would have been good to use an image (or object) of the type that is used in the evaluation later for better understanding.



**Time Spent Reviewing:**

2

---

> ### Author Response · Authors · 2021-08-10
> **To Reviewer qaZw**
>
> ### **References and Related Work**
>
> #### **Comparison with Listed References**
> > *Q: While working on vector graphics is not a very common research domain at the moment, the authors do not refer to the related work that has been looking at vector graphics*
>
> We thank the reviewer for the related works. We will cite the listed references and discuss the differences with our work in the related work section in the revised paper.
>
> We also want to highlight that our work is the ***the first to focus on conducting object detection directly from vector graphics without rendering them into pixels***.
> Although the works listed by the reviewer are related to our paper to some extent, they solve different problems (with different data format) and tasks.
> They cannot be applied to our recognition problems on vector graphics, so we believe they (though related) do not downgrade the contribution of our work. The summarization of the comparison between them are as follows:
>
> Following table is the summary of ***comparison between different works***:
>
> |   | Data type        | Task                          |
> | ------| ------------ | ------------------ |
> | Ours | Vector Graphics  | Recognition: Object Detection |
> | [1] | Vector Graphics  | Generation                    |
> | [2] | Handwriting      | Generation                    |
> | [3] | Handwriting      | Generation                    |
> | [4] | Handwriting      | Recognition                   |
> | BézierSketch[5] | Handwriting      | Generation: Handwriting to VG |
> | PlanIT[5]       | Semantic Objects | Planning                      |
> Note that ***existing Vector Graphics related works all focus on Generation***.  Apart from the task of generation, we believe recognition is a fundamental task and worth a dive into it.
>
> #### **Comparison with Handwriting**
> > *Q:  the entire domain of online handwriting and drawing recognition also handles vector graphics*
>
> Here is a summary of ***comparison between handwriting and vector graphics***:
>
> | Data type            | Structure                      | Primitive                                                  | Continuity | Resolution |
> | -------------------- | ------------------------------ | ---------------------------------------------------------- | ---------- | ---------- |
> | Vector Graphics      | Un-ordered                     | Shapes (in text) e.g. circle, rectangle | Yes        | Any        |
> | Handwriting (Online) | Spatially / Temporally Ordered | Discrete Points                                            | No         | Fixed      |
>
> We would like to conclude that ***handwriting and vector graphics are different data formats***. Online handwriting is essentially a variant of raster graphic or point cloud, which contains discrete points with temporal order. Handwriting does not have the merits of being scaled to other resolution without aliasing. The techniques on handwriting can not be trivially re-applied in vector graphics.
>
> To further demonstrate their differences. Here is the links of an example handwriting image file and vector graphic file.
> + Handwriting Sample: https://www.dropbox.com/s/e6kpyc6988sjz6g/handwritting_data_format.png?dl=0
>
> + Vector Graphics Sample: https://www.dropbox.com/s/c3ddeibliau5uu4/vg_data_format.png?dl=0
>
> #### **Comparison with Bézier Conversion in [4]**
> > *Q: [4] Fast Multi-language LSTM-based Online Handwriting Recognition - is using a Bezier representation - similar to what is proposed here for handling (vector graphics of) handwriting*
>
>   In the listed reference, [4] is the only one that also does a recognition task (but ***not object detection***).
>   Although ***two methods tackles totally different task*** (object detection on vector graphics vs. on-line handwriting recognition) and ***proposes totally different models***, [4] contains one component a bit similar to our work, which uses a sequence of Bézier Curves to fit a temporally ordered sequence of discrete pixels.
>
>   There are 3 major differences between ***Pixel-to-Bézier conversion*** and ***VectorGraphics-to-Bézier conversion***:
>   + ***[4] is not applicable to Vector Graphics.*** The Bézier curve sequence obtained in [4] is a temporally ordered sequential structure, and [4] uses a sequential model (RNNs) for recognition. It cannot be used in more general and un-ordered graph-like structures like Vector Graphics.
>
>   + ***Vector graphics give more precise Bézier conversion***. Converting discrete points into continuous curves is a process of cubic curve fitting. Hence  ***there is an inevitable trade-off between compactness of the representations and fitting errors***. A more compact and smooth curve sequence brings larger errors. On the other hand, vector graphics is compact by natural, and most of the conversions are costless. Only a small portion of the process brings a constant and negligible small error (e.g., less than 0.02% maximum radial error for circle).
>
>   + ***Vector graphics-to-Bézier Conversion is only a small part of our method.*** Besides being the first work to tackle recognition directly from Vector Graphics, YOLaT is a framework integrated with novel components including 1) a novel multi-graph construction method based on the Bézier Curve from the vector graphics, 2) a novel dual-stream GNN to extract feature representation from vector Graphics, and 3) a novel object detection head generating proposals and predicting object categories from vector graphics.
>
> ### **Details of Bézier curve conversion**
> > *Q: TECHNICAL in section 4, is a description of how the graph is build which includes a step in which the input drawing is converted to bezier curves. It is not clear however, how this would work in the case of a very long polyline or e.g. for a circle which are both not easily represented as a Bezier curve. It feels like for this the shape need to be split into multiple bezier splines - but it is unclear from the paper how that is done.*
>
>  How to convert shapes into Bézier curves is introduced in Section 5.1 Line 237-241 and our provided code shows the details of the conversion. If the curve can not be easily represented as a Bézier curve, we split the curve into several sub-parts so that they can be represented easily. For example, a circle is split equally into four arcs with equal radian and then converted in four Bézier curves. PolyLines or other intersected curves are split into several sub-curves, each represented as an independent Bézier curves sharing intersection points.
>  Detailed implementation can be found in the source code in the supplemental materials and will be added into the appendix in the revised paper.
>
> ### **Experimental Evaluation: Pre-training Influence and Dataset Scale**
> > *Q: EXPERIMENTAL EVALUATION.
> it is not surprising that a computer vision model that was pre-trained on very different data and then fine tuned on ~600 images doesn't perform particularly well on objects that are mostly defined by their outlines rather than by their textures.*
>
> > *Q: I personally don't think the evaluation in the paper is sufficient to understand limitations. Experiments on two datasets of 1000 images of unknown difficulty don't give a good idea of what the limitations of the method are.*
>
>  As shown in Table 1 and Table 2 in the manuscript, we compared the pixel-based models both with and without pre-training, and our model constantly out-perform both. The results also show that ImageNet pre-trained CNN actually performs better than its un-pre-trained counter part.
>
>  As for the dataset scale, in object detection, besides number of images, we should also look at the number of objects. Although the used datasets contain fewer images, high resolution Vector Graphics contain ***3 to 5 times more objects per image*** than normal object detection datasets like MS-COCO. As stated in section 5.2,  datasets used in our experiments contain totally 14,000 and 28,000 objects respectively, which is an acceptable size for an object detection task.
>
> ### **The meaning of 'Only Look at the Text' in the title**
> > *Q: Beyond that, I feel that the paper title and representation of the work is misleading: Just looking at the text would in my mind imagine just looking at the raw SVG file - however, as I am reading the paper it becomes clear that there is a lot of specialized preprocessing to normalize the vector graphics and create the graph. So, "only look at text" - is clearly not a good representation of the paper.*
>
>  In this paper, 'Only Look at the Text' in the title means that we propose a new object detection methods ***that does not need to render vector graphics*** into pixels as in most object detection papers do, and this is a method that performs recognition from vector graphics rather than pixels. Our pipeline includes the process of normalizing the vector graphics and creating the graph as mentioned by the reviewer. However, these processes are a part of our end-to-end pipeline and our model takes text format source file as our input. Please let us know if you still think this is a concern.

---

> > ### Comment · Reviewer_qaZw · 2021-08-17
> > **Response to authors reponse**
> >
> > Hi Authors,
> >
> > Thanks a lot for your careful response to my concerns. I still disagree on most of these.
> >
> > a) I still feel pretty strongly that the title is misleading in multiple dimensions. a) the method is not looking at the actual text _at all_ - this would in fact be a much harder problem in that it would require your method to understand the various different concepts in SVG graphics - but rather your method applies a "feature extraction" / re-encoding into a much more homogeneous and consistent format.
> > Further, the fact that this method is CNN-free is not actually the relevant part - there have been recent methods achieve state of the art performance using transformers and fully connected layers that didn't require a CNN either. If at all the relevant part of the method is that you operate without rendering the graphics to pixels - but using a manually engineered vector graphics representation (Bezier curves).
> >
> > b) I strongly disagree with the idea that online handwriting / sketch data is not vector data. I would argue that this type of data is vector graphics in a coherent format: polylines - and in fact using a Bezier representation for this type of data has done in at least one recent state of the art paper: https://arxiv.org/abs/1902.10525. Note also, how the method that you describe for the Bezier representation (when a line cannot be represented as a single curve, it is split into multiple parts) - sounds identical to the method presented in that paper. This paper is not working on pixels.
> >
> > c) I also still think that the experimental evaluation is massively lacking. I don't think the baselines are reasonable baselines. In fact, in the QuickDraw kaggle competition (https://www.kaggle.com/c/quickdraw-doodle-recognition) - which is online sketch recognition - I am pretty sure that some of the methods have used a combination of pixel- and online data (as per the discussion board).
> > So, I would definitely expect that a method that looks at vector data would look into some of the work that is happening in online handwriting  recognition. E.g., if you do online handwriting recognition with segmentation - you basically have an object detect system in a _sequential_ vector graphics. - all of that said - the data does NOT need to be considered sequential - as e.g. demonstrated in CoSE (https://arxiv.org/abs/2006.09930) - which models online drawing data as sets of strokes. Note that also one of the other reviewers suggests to try additional baselines using transformers - and e.g. the CoSE paper referred to above - is able to handle (handdrawn) vector drawing data with transformers (in a generative setting).
> >
> > I personally think this is a very important topic - handling of vector graphics is an important paper -  I am willing to up my review by 2 points because of that - for this particular paper, I still feel the execution - in particular wrt. experimental baselines -  is not sufficient.

---

> > > ### Author Response · Authors · 2021-08-18
> > > **2nd Response to Reviewer qaZw**
> > >
> > > We thank reviewer for once again taking extra time and efforts to give more insightful comments. We appreciate that the reviewer raises the score, acknowledges the importance our paper, and regards "handling of vector graphics" as "important paper".
> > >  We found reviewer's response to our response very helpful, and will further revise our paper accordingly.
> > >  Here are some some clarification to the reviewer's response:
> > >
> > >  ### **Handwriting as polylines.**
> > >  The reviewer provides a very insightful perspective that treats the ordered pixel sequence (or sparse samples of the dense pixel sequence) as coherent polylines. In a way, the coherent polylines can be seen as a ***special case of Vector Graphics*** where there are ***only one type of shapes***: Lines constrained by temporal order. However, a vector graphics format contains multiple types of un-ordered shapes with totally different attributes and properties, and hence need more general method.  Furthermore, existing polyline based hand-writing method focus on ***sequential recognition***, and cannot be used for ***un-sequential object detection***.
> > >  For clarity, we list the difference between coherent polylines and vector graphics in the following table:
> > >
> > > | Data type         | Structure          | Primitive                                     | Resolution | Task             |
> > > | ----------------- | ------------------ | --------------------------------------------- | ---------- | ---------------- |
> > > | Vector Graphics   | Un-ordered         | All Shapes e.g. circle, rectangle, arc, curve | Any        | Object Detection      |
> > > | Coherent Polyline | Temporally Ordered | Lines                                         | Fixed      | Recognition |
> > >
> > > As for the difference with the Bézier Conversion in https://arxiv.org/abs/1902.10525, we have listed the difference between our method and this paper in the previous response. Also Bézier Conversion is only a small part of our method, most of our method is totally different with this paper, including task (online handwriting recognition vs. object detection), model (LSTM vs. dual-path GNN), proposal generation methods and loss function.
> > >
> > > ### **Adding Non-sequential Hand-writing baselines**
> > >  We appreciate the reviewer's insightful suggestion to leverage the works in online handwriting like CoSE and excited to work on this topic  in the future work. Although we do not directly modify a specific hand-writing method into a baseline (it is really hard to do as explained later), methods with similar designs and core ideas have been compared:
> > >
> > > ***CNNs in Kaggle 1st Solution.*** From the results on the competition mentioned by the reviewer, CNN seems to be the SOTA un-sequential methods (even better than LSTM) in online handwriting. For example, we find that the 1st place solution achieves the best ***single model*** performance using CNN models (outperform LSTM). CNN based detector is already substantially compared in our paper.
> > > Furthermore, high performance solutions are ensemble of multiple models, while our paper focus on object detection with single model, and hence should not be compared.
> > >
> > > ***Transformer in CoSE.*** CoSE is a ***generation method*** taking pixel/point sequence as input, rather than vector graphics. A ***transformer*** is served as an encoder to extract stroke embedding, which is compared in three different ways in our paper (refer to the response to Reviewer jgFb).
> > > To directly design an object detection variants of CoSE, three ***major modifications*** are needed:
> > > 1. Design a proposal generation methods/model (e.g., anchor based proposals, rpn or other proposal generation methods).
> > > 2. Design a layer to aggregate individual stroke features into global feature of an entire object proposal. (e.g, Average Pooling, Transformer Layer)
> > > 3. Design object classification and offset regression loss for training.
> > >
> > > Note that all above modifications need carefully design and substantial experiments, almost leading to a new publish-able work.
> > >
> > > ***Other Transformer.*** Please refer to the response to Reviewer jgFb. We have experimented on three ways to apply transformer on vector graphics and all of them achieve worse performance than YOLaT.
> > >
> > > ***Handwriting Segmentation cannot be directly used for object detection.*** It is infeasible to use segmentation loss for object detection. Segmentation only classifies each primitive into a category, it is not able to decide which object the primitive belongs to. If you want an instance-level separation, the object detection technique has to be applied.
> > >
> > >
> > > ### **Issues with the Title**
> > > ***Only Look at Text.*** We understand the reviewer's concern on this part of the title. In our paper, 'Only Look at Text' mostly means 'Not Looking at Pixels'. We will add more explanation about this title in the revised paper, or take this phrase out of our title.
> > >
> > > ***CNN-Free.*** For this phrase, we do not see why it is misleading, because our proposed detector is indeed CNN-Free, just like Transformer. CNN-Free also address that our method is not based on pixels, while transformer methods like Detr and Swin-Transformer take pixels as input.

---

> > > > ### Comment · Reviewer_qaZw · 2021-08-20
> > > > **Further comments**
> > > >
> > > > ## Regarding representation:
> > > >
> > > > Two points:
> > > > 1. While it's true that SVG files can contain a large amount of distinct shapes - this is not relevant to the method here because all shapes are prenormalized to bezier splines - and thus the method does not actually know about different types of shapes
> > > >
> > > > 2. Sequentiality: An SVG file also is a sequence of shapes - and in many cases these are  contained in the order in which they were  added to the SVG file. Representing these as a set is a good idea - but it could be represented as a sequence (and on certain SVGs it might even help because there probably is information in the order of the sequence, e.g. I would imagine that items close together in space are also often close together in the sequence. Note, that CoSE uses a set representation of otherwise sequential data for the very same reason as you are pointing out.
> > > >
> > > > ## Title:
> > > > Here are two proposals:
> > > >
> > > > Recognizing Vector Graphics in Bezier Space
> > > > Recognizing Vector Graphics without rendering as pixels
> > > >
> > > > ## Baselines:
> > > > After more consideration, I acknowledge that it's infeasible to add a better baseline.
> > > >
> > > >
> > > > ## Finally:
> > > > So, I am willing to increase my score to "marginally above the acceptance threshold" now - assuming that
> > > > a) the authors acknowledge  in the paper that they don't consider these baselines strong baselines because the methods were designed for very different tasks  - yet, there are no stronger baselines readily available.
> > > > b) the title is changed

---

> > > > > ### Author Response · Authors · 2021-08-21
> > > > > **Third Response to Reviewer qaZw**
> > > > >
> > > > > We thank the reviewer for raising the score again. The reviewer's comments on the Bézier curve representation and sequentiality are very insightful and give us several interesting ideas on how to improve our model in the future work.
> > > > >
> > > > > As for the title, we promise to change it. Currently, we are leaning toward changing it to  "Recognizing Vector Graphics without Rasterisation", following the reviewers advise. Note that rasterisation literally means rendering as pixels.
> > > > >
> > > > > Finally, following reviewer's advice, in the revised paper, we promise to acknowledge that the baselines discussed in the response is not compared because they are designed for very different tasks and for now there are no stronger baselines readily available.

---

### Official Review · Reviewer_jgFb · 2021-07-12

**Rating:** 6
**Confidence:** 3

**Summary:**

The paper present a method to perform object detection for vector graphics images directly from their description as order three Bézier curves. It proceeds as follows:
1. it convert a vector graphics object as a set of order 3 Bézier cuvres
2. it looks at this set of curves as a graph
3. it process this graph using a GNN
4. it aggregates the features of the GNN on proposal regions, generated form the vector description of the image and classify them.

The method is evaluated on two public datasets and compared to standard CNN baselines on the rendered images.

**Ethics Review Area:**

["I don’t know"]

**Limitations And Societal Impact:**

limitations: it would be nice to discuss more the influence of the way the vector graphics was generated on the resulting Bézier curve-based representation as well as the errors made in the conversion.
societal impact: ok

**Main Review:**

I don't know of any other paper performing object detection in vector images directly from the vector description. If this is indeed true, I think it gives the paper enough novelty to be accepted despite the weaknesses listed bellow.

Weaknesses:
1. I would like a clear ablation showing the performance of CNN classifers (ResNet/Fast-RCNN) on the proposal extracted using the method from the paper + the proposed GNN using proposals from another method. Right now, it's unclear how much of the boost is coming from better proposals.
2. I found the technical part of the paper (part 4) very hard to parse. Notations could likely be improved, showing them on a picture might also help.
3. Beyond readability, I felt many of the choices were kind of arbitrary, even if several are justified by ablations. Assuming this is the first paper in this area, I think it's ok even if not ideal.
4. A transformer baseline/comparison would be very natural considering the structure of the data.
5. Why not perform segmentation instead of detection? To me it would be much more natural considering the data
6. It would be nice to include a quick literature review on vectorization, e.g. based on the related work of "Shen, I. C., & Chen, B. Y. (2021). ClipGen: A Deep Generative Model for Clipart Vectorization and Synthesis. IEEE Transactions on Visualization and Computer Graphics." or "Parakkat, A. D., Cani, M. P. R., & Singh, K. (2021, May). Color by numbers: Interactive structuring and vectorization of sketch imagery. In Proceedings of the 2021 CHI Conference on Human Factors in Computing Systems (pp. 1-11)."

misc.
- l 155 "an cubic"
- l 161-164 especially unclear
- section 3 seems completely useless to me

**Time Spent Reviewing:**

1.5

---

> ### Author Response · Authors · 2021-08-10
> **To Reviewer jgFb**
>
> ### **First work performing object detection in vector images directly from the vector description**
> > *Q: I don't know of any other paper performing object detection in vector images directly from the vector description. If this is indeed true, I think it gives the paper enough novelty to be accepted despite the weaknesses listed bellow.*
>
> Thanks for acknowledging our contribution.  To the best of our knowledge, this paper is indeed the first work that performs object detection in vector graphics directly from the vector description, or even the first recognition task on vector graphics.
>
> ### **More ablation study on YOLaT's proposal generation**
> > *Q: 1. I would like a clear ablation showing the performance of CNN classifers (ResNet/Fast-RCNN) on the proposal extracted using the method from the paper + the proposed GNN using proposals from another method. Right now, it's unclear how much of the boost is coming from better proposals.*
>
> The proposal generation is also one of our contributions. It is specifically designed for Object Detection on Vector Graphics, and creates much more precise and tighter bounding boxes than previous methods. Aside from our novel dual-stream GNN, this generation method also boosts the performance. As shown in the following table, by using our proposal generation method, the performance of FasterRCNN improves by around 4%, and applying YOLaT on the proposals further improves the performance.
>
> |                                                      | AP@50 on floorplan |
> | ---------------------------------------------------- | ------------------ |
> | Faster-RCNN-R18-FPN                                  | 80.9               |
> | Faster-RCNN-R18-FPN with YOLaT's proposal generation | 85.2               |
> | YOLaT                                                | 98.8               |
>
> ### **Section 4 is hard to follow**
> > *Q: 2. I found the technical part of the paper (part 4) very hard to parse. Notations could likely be improved, showing them on a picture might also help.*
>
> Thanks for the advise. We will adding more detailed explanation in section 4 and revise the illustration in Figure 2 and adding an extra figure to explain the proposal generation. One can also refer to our code for more details.
>
> ### **Some of the choices seems arbitrary**
> > *Q: 3. Beyond readability, I felt many of the choices were kind of arbitrary, even if several are justified by ablations. Assuming this is the first paper in this area, I think it's ok even if not ideal.*
>
> As the reviewer mentions, this is the first work in this area, and we believe recognition directly from vector graphics is an open problem.
> There are a lot of options waiting to be explored, such as transformer mentioned by the reviewer (please refer our response in the next question). We do not think our "Bézier Curves + Graph" is the ultimate solution for this task, and we wish that our work can inspire more researchers to work in this area to seek better solutions.
>
> ### **Transformer baseline**
> > *Q: 4. A transformer baseline/comparison would be very natural considering the structure of the data.*
>
> We have tried applying transformer in three different ways for this task, but didn't find superior performance.
> 1. ***Transformer takes the textual commands as sentence sequences*** as what typically done in natural language processing. This is definitely an intuitive way for this task, however, (possibly) due to the different nature of textual commands and natural language, lack of huge-scaled corpus like NLP datasets, this method does not give us satisfactory results.
> 2. ***Transformer takes shapes and corresponding parameters at inputs for relation calculations***. We've tried to use transformer and its attention mechanisms to group shapes into objects, and predict bounding boxes based on the group results. However, (possibly) due to the large number of shapes included in one image vector graphics, it is difficult for the attention mechanism to learn precise divisions of groups.
> 3. ***Transformer takes obtained Graph as input.*** As shown in table 3b, GAT, a self-attention based model taking graph as input is compared and found no improvements compared to a simpler version as in YOLaT.
>
> Again, recognition directly from Vector Graphics remains an open problem, and it is worth investigating on a better way to apply transformer and other advanced deep learning techniques in this task in the future.
>
> ### **Why not perform segmentation**
> > *Q: Why not perform segmentation instead of detection? To me it would be much more natural considering the data*
>
> The existing public datasets only provide bounding box annotation for object detection, and do not have per-pixel/primitive annotation of shape categories.
> Furthermore, segmentation only classifies each primitive into a category, it is not able to decide which object the primitive belongs to.
> Even most instance-level segmentation methods require object detection before conducting pixel/primitive-level classification.
>
> ### **Adding related work on generation and vectorization**
> > *Q: It would be nice to include a quick literature review on vectorization, e.g. based on the related work of "Shen, I. C., & Chen, B. Y. (2021). ClipGen: A Deep Generative Model for Clipart Vectorization and Synthesis. IEEE Transactions on Visualization and Computer Graphics." or "Parakkat, A. D., Cani, M. P. R., & Singh, K. (2021, May). Color by numbers: Interactive structuring and vectorization of sketch imagery. In Proceedings of the 2021 CHI Conference on Human Factors in Computing Systems (pp. 1-11)."*
>
> Thanks for the advice, these reference will be added to the revised paper. The listed references all focus on vector graphic ***generation***, and we hope our work could draw more attention  on vector graphics based ***recognition***, which is a fundamental CV task.
>
> ###  **Bézier curves conversion error**
> > *Q:  it would be nice to discuss more the influence of the way the vector graphics was generated on the resulting Bézier curve-based representation as well as the errors made in the conversion.*
>
> The conversion of most primitives are lossless, such as lines, polygons.
> For some primitives like Circle and Arc, the conversion error is extremely small. For example, as stated in the response for Reviewer z9Ww, the maximum radial error for converting Circle to Bézier curve is less than 0.02%. Compared to the error from converting vector graphics to pixels, this error can be regarded as negligible.

---

### Official Review · Reviewer_NPkQ · 2021-07-14

**Rating:** 6
**Confidence:** 4

**Summary:**

In this paper, the authors proposed one framework for the object detection in the vector graphics instead of raster graphics (i.e., pixel images), where the main contribution maybe is to employ the graph neural networks in this area. The extensive experiments on two datasets show the effectiveness of this method, especially comparing with the methods on raster graphics. However, I do not think this comparison is fair, since the proposed methods employ more information contained in the vector graphics while other methods only use pixel information.

**Limitations And Societal Impact:**

I do not find any negative societal impact, and the suggestion are in the box of Main review.

**Main Review:**

The work in this paper is for the vector graphics, which is interesting. One problem is that how to utilize the detailed information in the vector graphics.

In this paper, the authors transforms all the primitives into a unified format, where the Bézier Curve is used, then graph neural networks are employed for the object detection. The experimental results are good, and I think this work is significant for the research on vector graphics.

However, some details are better to give:

(1) In the Eq. (4), based on my understanding, there are several points P between the two points of this edge, so the x^e is obtained by concatenating all of these points?

(2) It is better to give more details on how to get the regional cluster C_k, since this plays an important role in the following steps.

(3) In the Eq. (7), are all of the z_i in the same cluster C_k the same? My understanding is that neighbors in the same cluster are the same (i.e., N_i^p) because they are dense connections. Please correct me if my understanding is wrong.

(4) In the table 3b, it is not clear that the aggregation functions (last three rows) are replaced by three popular GNN methods in stroke-wise stream, or position-wise stream, or both. Moreover, why the performance of three popular GNN aggregation methods are reduced largely compared to the proposed two aggregation functions? BTW, in the second part (position-wise stream), I guess the h_j - h_i may be z_j - z_i because there is no variable h in this stream.

(5) In the proposal generation, the authors mentioned that "permute all vertex pairs on the grid mesh ... rectange region". does it mean every two nodes in the C_k will get one rectangle region? if it does, there will be a large number of rectangle regions.


**Time Spent Reviewing:**

8

---

> ### Author Response · Authors · 2021-08-10
> **To Reviewer NPkQ**
>
> ### **How to obtain $x^e$ in Eq.(4)**
> > *Q: (1) In the Eq. (4), based on my understanding, there are several points P between the two points of this edge, so the x^e is obtained by concatenating all of these points?*
>
> As introduced in section 4.1, we use a cubic Bézier Curves to represent the primitives, and hence $x^e$ is always the concatenation of the coordinates of two off-curve control points.
>
> ### **Details on obtaining the regional cluster $\mathbb{C}_k$**
> > *Q: (2) It is better to give more details on how to get the regional cluster C_k, since this plays an important role in the following steps.*
>
> Thanks for the advice and more details on this will be added to the revised paper (or please refer to our code). The detailed steps of generating $\mathbb{C}_k$ is as follows:
> + Given our graph representation of a Vector Graphic, we first obtain all the connected components in the graph, based on the stroke-wise edges $\mathcal{E}_s$.
> + For each pair of connected components, obtain their expanded minimum bounding rectangles and the overlapping area of the rectangles.
> + If the expanded area of two connected components overlap, merge two connected components into one $\mathbb{C}_k$.
>
> ### **Are all $z_i$ in the same cluster $\mathbb{C}_k$ the same**
> > *Q: (3) In the Eq. (7), are all of the z_i in the same cluster C_k the same? My understanding is that neighbors in the same cluster are the same (i.e., N_i^p) because they are dense connections. Please correct me if my understanding is wrong.*
>
> You are correct and we design Eq.7 in this way intentionally. As stated in Line 193-195 and Line 231-233, since all the nodes in one $\mathbb{C}_k$ shares the same $z$, Eq.7 only needs to be computed once for each $\mathbb{C}_k$, which significantly reduces the computational cost. The ablation study in Table 3 shows this design reduces the computational cost and has no negative effects on the model performance.
>
> ### **Ablation in Table 3**
> > *Q: (4) In the table 3b, it is not clear that the aggregation functions (last three rows) are replaced by three popular GNN methods in stroke-wise stream, or position-wise stream, or both. Moreover, why the performance of three popular GNN aggregation methods are reduced largely compared to the proposed two aggregation functions? BTW, in the second part (position-wise stream), I guess the h_j - h_i may be z_j - z_i because there is no variable h in this stream.*
>
> The aggregation functions in both stroke-wise and position-wise stream are replaced.
>
> YOLaT outperforms other GNN layers because our aggregation function has some novel and specific designs for the multi-graph generated from Vector Graphics. For example, our stroke-wise stream aggregation function in Eq.6 considers both the feature of neighboring nodes themselves and the feature difference between neighboring nodes. We add extra BN in the transformation function in $f^s$ and $f^p$ to normalize the feature distribution. Like GAT, we also choose a no-linear transformation function to encode node feature, which is why GAT also outperforms GCN and GraphSage.
>
> Thanks for pointing out the typo. The reviewer is correct about $h_j - h_i$ should $z_j - z_i$ in the second part of the table, which will be updated in the revised paper.
>
> ### **Permuting vertex pairs results in too many proposals**
> > *Q: (5) In the proposal generation, the authors mentioned that "permute all vertex pairs on the grid mesh ... rectange region". does it mean every two nodes in the C_k will get one rectangle region? if it does, there will be a large number of rectangle regions.*
>
> The reviewer's understanding on the proposal generation is correct. However, the average number of proposals in YOLat is around $500$, which is much smaller than RCNN variants.
> The number of proposals is $O(NM^2)$ where $N$ is the number of $\mathbb{C_k}$, and $M$ is the number of vertex in a grid mesh. In our implementation, mean $N$ is around $25$ and $M$ is set to $6 \times 6$, we also filter out invalid proposals such as ones with extremely small size or ones containing fewer than 3 nodes.

---

### Official Review · Reviewer_z9Ww · 2021-07-16

**Rating:** 6
**Confidence:** 4

**Summary:**

The paper addresses the problem of object recognition, i.e., object localization and classification, in Vector Graphics.
The input is a text document that describes the vector graphics, and the output is a bounding box corresponding to a localized object.

The proposed approach is supervised.

The main idea is to represent the input textual document describing the Vector Graphic (VG) object using a unified representation for different primitives described in the text document (such as a Bezier curve). The VG objects are represented using a Graph, where the nodes correspond to the start and end points of the Bezier curves.

A dual-stream Graph Neural Network (GNN) is proposed to aggregate feature representation for the input VG object, which is in-turn used for classification and localization.

The method is evaluated using Average Precision metric.

Two kinds of publicly available VG datasets are used: Floorplan and Diagrams.
Comparison against object recognition techniques that employ a CNN has been done, and the efficiency
of dual-stream GNN architecture is demonstrated.

**Limitations And Societal Impact:**

Yes, the authors have adequately addressed the limitations and potential negative societal impact of their work.

**Main Review:**

Originality:

I think the paper is addressing an interesting problem of object recognition in Vector Graphics input. The proposed idea, especially extracting the textual data into Bezier curves, is good. I have one major question here:
1) The VG input is in text format. How do you extract Bezier curves from there? In other words, how do you parse the input textual document to understand what kind of primitives are described in there?

There needs to be more clarity about the "Format Unification" step shown in figure 2.

Quality:

The proposed method has shown to significantly outperform object recognition methods that employ a CNN.
Since no CNN is made use of in the proposed method, the model is relatively light-weight and yet, efficient.
In terms of writing and mathematical formulation, the paper is reasonably written.

Clarity:

The flow of the paper is easy to follow. I would suggest the following to help improve the exposition of the paper even more:
State clearly what is the input. I think this is where the readers need to be educated. Let them know what VG format is and how is it described in general. Also describe how such a textual document of VG is parsed into a data structure that is fed to the employed neural network.
Explain how the proposal generator works. I am still not very clear about the proposal generator. I think Section 4.3 can be broken down in a better way. Perhaps start of with a different paragraph about proposal generators and then describe the training loss.
And finally, be specific about the representation of the output.


Significance:

The problem being addressed is important. Dependence on raster formats of data is both computationally heavy and resolution-constrained. Computational methods that can operate on VG data and perform visual recognition tasks as that of raster form of data are much needed and perhaps will become dominant in the next few years.

**Time Spent Reviewing:**

2 hours

---

> ### Author Response · Authors · 2021-08-10
> **To Reviewer z9Ww**
>
> ### **How to extract Bézier curves**
> > *Q: The VG input is in text format. How do you extract Bezier curves from there? In other words, how do you parse the input textual document to understand what kind of primitives are described in there?*
>
> A vector graphic file contains multiple lines of textual commands, and each line defines a specific primitive/shape in the image, including the shape type (e.g., circle, line, Bézier curve, etc) and its associated parameters (e.g., start/end/center point coordinates).
>
> After parsing the shape category and parameters from the command, each shape (or a part of the shape) can be converted into a Bézier curve with a closed-formed expression.
> Here we take circle as an example.
> A circle is split into four equal sections, ie., left-up quarter, left-bottom quarter, right-up quarter and right-bottom quarter, and each is converted to a Bézier curve.
> For a circle centered at the origin with radius 1, and the right-up quarter start at (0, 1) and end at (1, 0), the control points of the corresponding Bézier curve can be obtained by:
> $$
> P_0=(0,1), P_1=(c,1), P_2=(1,c), P_3=(1,0), c=0.551915024494
> $$
> which gives a maximum radial error to the original circle less than 0.02%.
>
> Due to page limit, we only briefly introduce it in Section 5.1 Line 237-241. More details can be found in the source code in the supplemental materials and will be added into the appendix.
>
> ### **Clarity of the paper**
> > *Q: The flow of the paper is easy to follow. I would suggest the following to help improve the exposition of the paper even more: State clearly what is the input. I think this is where the readers need to be educated. Let them know what VG format is and how is it described in general. Also describe how such a textual document of VG is parsed into a data structure that is fed to the employed neural network. Explain how the proposal generator works. I am still not very clear about the proposal generator. I think Section 4.3 can be broken down in a better way. Perhaps start of with a different paragraph about proposal generators and then describe the training loss. And finally, be specific about the representation of the output.*
>
> We thank the suggestion of the reviewer. In the revised paper, we will expand section 4.1 and give a more detailed description on what is vector graphics, and how vector graphics are parsed and converted to Bézier Curve. We will set up a separate sub-section to introduce how the proposal generation works. Our code is also included and provide the details.
> For a better clarification on the proposal generation, the process can be broken into following steps:
> + Evenly slices each regional clusters into grids.
> + Permute all vertex pairs on the grid mesh, each of which forms the top-left and bottom-right points of a rectangle region.
> + The nodes, edges and corresponding primitives within the rectangle region forms a proposed object.
> + Obtain the minimum bounding rectangle of the shapes within the rectangle region to form the bounding box of an object proposal.

---

> > ### Comment · Reviewer_z9Ww · 2021-08-18
> > **Response to Authors' response**
> >
> > Dear Authors,
> >
> > Thank you for responding to my questions, and providing clarifications.
> >
> > Here are some things I would like to bring to your notice:
> >
> >
> > 1) The title of the paper is poorly chosen. I think you have heard enough of this from other reviewers, so I won't go over this again.
> >
> > 2) As R-jgFb said, section 3 is completely useless. This space could instead be used for providing distinctions to text-based works, and if possible, incorporating the different tables presented in response to different reviewers.
> >
> >
> >
> > I lower my score from a 7 to a 6, mainly after reading through other reviewers' concerns.
> > This is not a negative sign for the paper, but a fair rating provided based on different aspects of the work (and the conference standards).

---

> > > ### Author Response · Authors · 2021-08-19
> > > **Second Response to Reviewer z9Ww**
> > >
> > > Thanks again for the reviewer's valuable suggestions.
> > > 1. We are working on a better title. We are considering taking out the phrase "You Only Look At Text" or change it to "You Only Look At Curves".
> > > 2. The goal of section 3 is to give a detailed explanation and clarification on the scope and the problem we focus on, namely the object detection on vector graphics. Following reviewer's suggestion, we will merge some of the content into section 4, and adding more comparison with other works and data formats like handwriting  (including tables shown in the responses) into section 2.

---

### Author Response · Authors · 2021-08-10
**To All Reviewers**

# **To All Reviewers**

We would like to thank all reviewers for your time and efforts in reviewing and providing insightful and constructive feedback to our paper. We are encouraged that you found our work important (reviewer z9Ww), interesting and significant (reviewer NPkQ).
We would also like to highlight that to the best of our knowledge, our work is the first to address the problem of recognition ***directly*** from vector graphics.

We believe that vector graphics based recognition is a new research area with great potential and wide applications. Currently it is still an open challenge, and our approach although not perfect may inspire more research efforts in this emerging area.
Below we provide detailed responses to each reviewer's comments to clarify the misunderstandings, and we will incorporate these clarifications in the final version if accepted.

---

### Decision · Program_Chairs · 2021-09-27

**Decision:**

Accept (Poster)

**Comment:**

The reviewers have extensively discussed the paper with the authors and came to the conclusion that the paper studies an interesting and important problem. The reviewers have agreed to support the acceptance of the paper if some of the changes are incorporated in the final version. Specifically, the title should be changed to be more precise and less misleading, and there should be more discussion on the feasibility of potential baselines (including the ones coming from handwriting recognition).
Overall, I agree with the reviewers and recommend acceptance of the paper but I would like to encourage the authors to carefully incorporate all the raised concerns and suggestions to improve the paper.